# Neighborhood level chronic respiratory disease prevalence estimation using search query data

**Nabeel Abdur Rehman**[1], **Scott Counts**[2]*

**1** New York University, New York, NY, United States of America, **2** Microsoft Research, Redmond, WA, United States of America

* counts@microsoft.com

## Abstract

Estimation of disease prevalence at sub-city neighborhood scale allows early and targeted interventions that can help save lives and reduce public health burdens. However, the cost-prohibitive nature of highly localized data collection and sparsity of representative signals, has made it challenging to identify neighborhood scale prevalence of disease. To overcome this challenge, we utilize alternative data sources, which are both less sparse and representative of localized disease prevalence: using query data from a large commercial search engine, we identify the prevalence of respiratory illness in the United States, localized to census tract geographic granularity. Focusing on asthma and Chronic Obstructive Pulmonary Disease (COPD), we construct a set of features based on searches for symptoms, medications, and disease-related information, and use these to identify illness rates in more than 23 thousand tracts in 500 cities across the United States. Out of sample model estimates from search data alone correlate with ground truth illness rate estimates from the CDC at 0.69 to 0.76, with simple additions to these models raising those correlations to as high as 0.84. We then show that in practice search query data can be added to other relevant data such as census or land cover data to boost results, with models that incorporate all data sources correlating with ground truth data at 0.91 for asthma and 0.88 for COPD.

## Introduction

Respiratory illnesses are among the most common chronic health issues, negatively impacting the day-to-day lives of millions of people around the world, while placing substantial burden on healthcare systems [1]. In the United States alone, more than 25 million people currently are diagnosed with asthma [2], and more than 15.7 million people suffer from Chronic Obstructive Pulmonary Disease (COPD) [3]. Chronic lower respiratory diseases, including asthma and COPD, are also a leading cause of death in the USA [4]. To date, no cure exists for COPD or asthma, though with proper medication and preventative measures, manifestation of severe symptoms and progression of these diseases can be controlled. In addition to already high mortality rates of these respiratory illnesses, individuals with pre-existing respiratory

**Data Availability Statement:** Census data is publicly available at the US Census Tract Bureau website. Census tract level ground truth CDC data is publicly available at 500 Cities project website. Landcover data is publicly available at the National

Landcover Database website. For privacy reasons, search engine companies cannot, and should not, ever share raw query data with the public. Thus, search query data, used to train models, cannot be publicly made available. However, minimal dataset, including computed landscape features, and census tract level predictions from our models used in evaluation metrics and generating plots, will be made available at: https://github.com/microsoft/UrbanInnovation-Dataset.

**Funding:** Microsoft Research provided support in the form of salary for author SC, but did not have any additional role in the study design, data collection and analysis, decision to publish, or preparation of the manuscript.

**Competing interests:** SC is affiliated with Microsoft Research. This affiliation does not alter our adherence to PLOS ONE policies on sharing data and materials.

conditions such as asthma and COPD, have an increased risk of severe illness from other diseases such as coronavirus (COVID-19) [5].

An important component of these preventative measures is understanding the integral role environmental and personal care factors play in the manifestation of respiratory illness symptoms. For instance, smoking tobacco and long term exposure to pollution, chemicals and fumes are known to cause COPD [6–8], while factors such as dust, air pollution, pollen count, and stress can trigger severe attacks in asthma patients [9].

Monitoring and reducing these environmental factors requires considerable effort on the part of governments. Critical factors such as air quality and dust may be localized to small scale areas, varying substantially based on their distance from industrial and high traffic areas, for instance [10]. Because local governments often have limited resources to devote to such monitoring and mitigation efforts, strategies and technologies that can help governments prioritize resource allocation to areas of greatest need are of crucial value [11]. In particular, identify geographic areas, as granular as neighborhood scale, where people are either at a higher risk or already suffer from respiratory illnesses, can help governments more efficiently and effectively address this resource allocation challenge.

The recent deadly outbreak of COVID-19 has cast a harsh light on the need for such risk identification, as individuals with preexisting conditions tend to have more severe symptoms and higher mortality rates than do healthy individuals. While asthma and COPD are themselves not known to be transmitted from human-to-human, these disparate COVID-19 outcomes illustrate that other infectious diseases that are highly contagious can increase the mortality rate of individuals suffering from these respiratory illnesses. Thus, given the extent of the COVID-19 outbreak, and the potential of similar outbreaks in future, there is a dire need to identify these 'high risk' groups at a neighborhood scale so appropriate measures could be taken to isolate them and reduce their contact with infected individuals.

The best data source to date for such geographically granular risk identification in the U.S. comes from the Centers for Disease Control and Prevention (CDC). In 2015, the CDC and the Robert Wood Johnson Foundation started a partnership to collect small area estimates of 27 chronic disease measures, including respiratory illnesses, at census-tract level, in 500 major cities in USA [12]. Census tracts are the smallest geographical boundaries available in the U.S. for which population, socioeconomic, and demographic data is collected. With an average population of roughly 4,000 inhabitants, and with nearly homogeneous socioeconomic and demographic factors inside each tract [13], these spatial units provide an ideal resolution to measure the prevalence of diseases.

These CDC estimates are based on survey data, and while they are invaluable in identifying respiratory illness rates at geographic scales small enough to support targeted mitigation strategies, unfortunately they are costly and difficult to carry out at frequent time intervals. Critically, these surveys also face sampling coverage challenges, forcing the CDC to extrapolate from the small number of sampled individuals in surveys using census data based on demographic similarity [14].

In light of both the potential public health benefits and the challenges with existing measurement techniques, we examine the applicability of search engine query data to generate census tract level estimates of the prevalence of respiratory illnesses. The intuition is that people suffering from a respiratory illness are likely to search for information about the illness, such as symptoms and medications and other treatments, and thus the number of people issuing such searches in a given area may be a good indicator of the percent of the local population affected by the illness [15, 16]. To align with the CDC survey, we test this approach on asthma and COPD, and at a census tract geographic scale. To highlight the fine spatial granularity of this geographic scale, there are more than 74,000 census tracts in the U.S. [17], of which we

included over 23,000 tracts, all from urban areas, which are more likely to contain the afore-mentioned environmental triggers of respiratory illnesses. To the extent search query data add value, we envision a process by which search companies make available anonymous, aggre-gated, and actionable indices of illness rate estimates to public health researchers and officials.

We first discuss methodological challenges to such an approach, including identifying rele-vant search queries from a large corpus of search query data, and resolving sparse feature val-ues given the extremely fine spatial granularity. Our results then show that models using search query data alone can generate out of sample respiratory illness rates that correlate with CDC illness rate estimates in the 0.7 to 0.8 range. We then demonstrate how a practitioner might utilize these search query-based models by incorporating data sources that capture envi-ronmental and social factors to yield even stronger estimates in practice. Final best models that combine search query, landscape, and demographic data, generate out of sample estimates that correlate at up to 0.91 with CDC estimates.

An additional benefit of our approach is that it could be extended to estimate the prevalence of respiratory concerns across the entire U.S., including cities not part of the 500 cities included in the CDC survey, simply by sampling query data from more geographic areas. Thus, the near-complete geographic coverage of query data obviate the need for extrapolation of estimates to non-sampled areas. Further, the ongoing nature of search query data could also support more frequent updating of respiratory illness estimates, providing for instance quar-terly illness rate estimate updates in order to better capture seasonal or evolving demographic and urban development changes.

The primary contributions of this work are:

- Demonstrate that search query data can generate localized sub-city census tract level esti-mates of asthma and COPD.

- Outline a method for search query keyword selection, for generating census tract level esti-mates, given data sparsity.

- Show how query data can be combined with other data (satellite, census) to generate even stronger illness rate estimates in practice.

## Related work

### Search query data for disease prediction

Search query data have been used for disease prediction, though most previous work has focused on prediction at state or city geographic scale. Google Flu Trends [16], perhaps the most well-known such work, divided the U.S. into 10 regions and provided weekly estimates of influenza like illness (ILI) at this region level. Subsequent work has studied the utility of search query data at country, state, county and city level for prediction of dengue, influenza, kidney stones, and lyme disease [18–21]. Together this body of work hints at the utility of search query data for disease prediction, yet it is limited in practical utility due to the large spa-tial units for which their predictions are made. To ensure that the predictions provide esti-mates at actionable spatial granularity for government officials and policy makers, there is a need to reduce the size of the spatial units for which estimates are generated [11].

Most previous work has used search query data for the prediction of diseases that are sea-sonal [16, 18, 20, 21], and thus can leverage a temporal component for feature creation and time series modeling. Because these disease are short lived, there is a higher likelihood of peo-ple searching disease related terms as opposed to long term diseases, for which people don't regularly search related terms. In the present study, we aim to estimate the geographical

distribution of non-seasonal, non-infectious long term diseases, which thus provide an added challenge of data sparsity in search volumes.

## Respiratory illness prediction

Many prior studies have explored the presence or absence of asthma and COPD in individuals based on the symptoms reported in their clinical visits. A systematic review of these studies is presented in [22], which stresses the need for improved tests and bio-indicators for identification of asthma in clinical patients. Given the cost of such tests, this review also calls for the need to identify sub-groups of individuals who are at high risk of asthma development. These guidelines fall in line with our goal of identification of respiratory illnesses at small geographical units for better allocation of resources.

Other work in this domain includes survey based studies that identify the association between various sociodemographic factors and respiratory illnesses. Ethnic background and parental history of asthma are known to have an association with increased likelihood of asthma in children [23]. While such studies do not provide estimates of prevalence of respiratory illnesses, they provide valuable information on factors associated with the prevalence of asthma in a population.

## Sub-city scale illness rate estimation

With the increased focus on improving the allocation of resources to reduce the burden of diseases, numerous studies have explored the relationship between alternative data sources and outcome data at a sub-city local scale. Maharana et al. used a deep learning framework to identify high dimensional embeddings from satellite imagery, to estimate census tract level prevalence of obesity in 5 cities in U.S [24]. The work provides valuable insights into the predictive capabilities of satellite imagery data, but given the smaller scope of the study (5 cities) and given that the high dimensional embeddings do not represent actual resolved land cover features, it is difficult to understand the association between land cover features and obesity. Nonetheless, land cover data are quite relevant to the study of respiratory illness, and thus for our work we show how pre-classified land cover features can be combined with features from search query data to improve predictions in practice.

Perhaps the most related work to our study is presented in [25]. Leas et al. explored the relationship between Chronic Heart Diseases (CHD), COPD and asthma, demographic features, smoking prevalence, and the number of tobacco retailers, at a census tract level in 500 cities in the U.S. The authors found a strong relationship between smoking and chronic diseases. Additionally, the authors found that lower median household income, smaller non-Hispanic white population and increased number of tobacco retailers was related with increased prevalence of smoking. Our work aims to augment and provide an alternative to survey datasets by leveraging search query data that are nearly complete geographically and may surface information not captured in surveys, such as perceived symptoms and unrealized environmental triggers.

# Materials and methods

## Datasets

**CDC data.** Census tract level estimates of 27 chronic disease measures for 500 major cities in the U.S. are publicly available on the 500 Cities project website [12]. The project combines geo-located health surveys from the Behavioral Risk Factor Surveillance System (BRFSS) with local demographic information from census data, using a multi-level regression, to provide census tract level estimates of chronic diseases [14]. BRFSS surveys are conducted over the

phone by the health department of each state, and are funded by the federal government. The methodology identifies location and demographic specific priors of each chronic disease and then uses the census data to extrapolate the estimates of chronic illnesses at a census tract level. For our study, we use the 2016 estimates of asthma and COPD, and despite any sampling limitations, consider them to be ground truth.

**Search query data.** We received a sample of approximately 11.5 billion anonymized search queries from Microsoft's Bing search engine from the year 2018 (data compliance statement provided in "Search Query Feature Selection" subsection). Based on the provided information, each search query was geo-located using primarily a reverse IP lookup. Search queries were filtered based on a keyword dictionary of terms related to asthma and COPD symptoms and medications (see Table 1 for examples and below for details on how these keywords were determined). These terms were selected for high precision matching to the diseases in question, and represent only a tiny fraction of all searches, yielding approximately 19.7 million disease-relevant queries. Using the geo-location information of search queries and the boundaries of census tracts, we assigned each search query to a census tract. For each relevant search query term, we calculated normalized counts for each census tract, dividing the count of search queries containing the term generated from a census tract by the total count of search queries generated from the census tract.

$$S_i(k) = \frac{|Q_i \supset k|}{|Q_i|}$$

where $S_i(k)$ represents the normalized count for search keyword $k$ in census tract $i$ and $Q_i$ represents the set of search queries generated from census tract $i$. This normalized count of queries containing disease-relevant keywords serves as the primary measure for each census tract.

In regards to the accuracy of reverse IP method to assign a query to a census tract, we note here that the likelihood of a search query generated from the center of a census tract belonging to the correct census tract is higher compared to queries generated closer to the boundaries of the census tracts. To reduce potential errors in reverse IP lookup of queries closer to census tract boundaries, we perform smoothing as described in "Census Tract and Neighboring Tract Selection" sub-section. Further, we hope advancements in location tagging and geo-location services in future can further help reduce such errors.

Additionally, we note here that the search query data available to us from the search engine was two years more recent than the ground truth CDC data (2018 instead of 2016). Given that we are concerned with geographic distribution rather than temporal forecasting, the two year time difference between the datasets in fact penalizes the query data to the extent that the population and/or environment changed non-uniformly across census tracts from when the CDC data were collected. To the extent both data sources reflect underlying ground truth, a contemporaneous query dataset should align more strongly with CDC data. Thus, if anything this temporal discrepancy would underestimate the predictive performance of search query data. Further, previous studies, relying on alternative data sources have also similarly used more recent data compared to ground truth [26].

**Table 1. Subset of keywords used to generate estimates of respiratory illnesses.**

| Keyword Type | Identification Method | Search Keyword |
|---|---|---|
| Related Symptoms | Word Embedding | cough, wheezing, migraine, anxiety |
| Related Diseases | Word Embedding | asthma, copd, bronchitis, pneumonia, sepsis, sinus, emphysema |
| Medication | Medical Literature | inhaler, advair, symbicort, ventolin, proair |
| Smoking Related | Word Embedding | cigarette, nicotine, cigs, juul, vape |

**Land cover data.** As mentioned, to demonstrate practical usage scenarios in which search query data is combined with other data to maximize illness rate estimation, we include results that incorporate land cover data from satellite imagery. The U.S. Geological Survey (USGS), in partnership with several federal agencies publishes the National Land Cover Database (NLCD), which provides country wide land cover classification of 30 m by 30 m satellite imagery for the U.S. [27]. This dataset is publicly available and consists of 8 main land cover classes and 20 sub-classes of land cover. These classes include Water (open water and perennial ice/snow), Developed (high, medium, low and open space), Barren (rock/sand/clay), Forest (deciduous, evergreen and mixed), Shrubland (dwarf shrub and shrub), Herbaceous (grassland, sedge, lichens and moss), Planted/ Cultivated (pasture/hay and cultivated crops) and Wetlands (woody and emergent herbaceous). The land cover maps are published every few years and are back dated (land cover maps of 2016 were published in early 2019). We use NLCD data for 2016, which coincided with the estimates from CDC.

In contrast to some recent work with satellite imagery that use deep neural networks to extract segmentation maps of land cover, we opted to instead use pre-classified maps from NLCD. The decision was made to ensure that we had reliable estimates of land cover classes in the absence of data for training image segmentation models [28].

**Census data.** For our final data source we use demographics exacted from census data. This data is used first to establish a target accuracy for illness prediction, and then in a final best prediction model that incorporates all three of search query, land cover, and census data. Census data provides a strong accuracy target in this case first because demographics correlate with respiratory illness related behaviors generally (e.g., smoking) [29], and more directly because the data was used by the CDC to help generate disease estimates in the 500 Cities project, which is our ground truth dataset. Census data was retrieved from US Census Bureau API [30]. Given the small size of census tracts, the Census Bureau publishes 5 year averages of census data for census tracts. For each census tract, we retrieve 20 demographic indicators representing age groups (9 features), gender (2 feature), race (8 features) and poverty level (1 feature) and calculate the proportion of population pertaining to each feature. The poverty level feature was computed by identifying the percentage of population of a census tract living below poverty line.

## Search query feature selection

As mentioned above, we identified a set of high precision keywords that could in turn be used to identify illness-related search queries. As a first step in that process, we trained a word embedding model on a 1% sub-sample of the queries in order to identify terms that appear in similar search query contexts to one or more of three primary illness terms: 'asthma', 'copd' and 'smoking'. Although smoking was not directly one of our illnesses, 'smoking' was included given that it is the leading cause of COPD [31].

For our word embedding model, we use a Skip-Gram Word2Vec model [32]. Skip-Gram models are single hidden layer neural network that convert text words into a $d$ dimensional vector representation, based on the occurrence of words in the window $w$ around them, i.e the context of the word. Thus, words with similar context have similar vector representations. For our work we used the value of $d = 100$, which is commonly used in Skip-Gram models and a window of $w = 6$, which is a typical word length of a search query. After training this model, we sorted the list of all unigrams based on cosine similarity to the illness terms in the word embedding vector space, and selected for inclusion as keywords all terms until the occurrence of first non-disease related term.

Additionally, given that patients are likely to search for related medicine, dosage and side effects, from the medical literature we added a list of common medication names used by asthma and COPD patients. Between the two sources, a total of 88 keywords were identified (see Table 1 for examples). Given the high spatial granularity of the study, the addition of related terms from medical literature allowed us to capture any sparse signals which could be representative of disease prevalence. Given the large overlap in symptoms and medication of both asthma and COPD, we used the same set of keywords when predicting the prevalence of both illnesses and assumed that any variations in the predictive capabilities of the keywords would be accounted for in our model.

We then take two additional steps to further address any data sparsity within a given census tract. First, we introduce additional features for our model that represent the sum of i) all related symptom queries, ii) all related disease queries, and iii) all related medication queries. Second, for each census tract $i$ we identified $m - 1$ closest neighbouring census tracts ($m^{th}$ census tract is itself) and take the average of normalized counts of feature across those $m$ census tracts.

$$\hat{S}_i(k) = \sum_{i=1}^{i=m} S_i(k)/m$$

where $\hat{S}_i(k)$ represents the average value for search keyword $k$ for census tract $i$ to be used in the model. $S_i(k)$ represents the actual value of search term $k$ for census tract $i$. The rationale for this aggregation is that it both accommodates any small amount of error in the query geolocation and also incorporates real-world behavior in which people who live in a given census tract are likely to walk, drive, shop, bike, etc. in adjacent tracts (recall that census tracts generally are small, roughly neighborhood size areas).

Finally, with respect to data privacy, the sample of queries made available to us was completely anonymous, and in fact other than manual inspection of sub-samples during feature creation, the query text itself was never exposed. All analyses were performed on aggregates, both geographically (census tract) and temporally (2018), and thus no individual's data was ever isolated in any way. Any census tract that contained data generated by fewer than 50 unique people was removed from study to further preserve privacy, though the vast majority of tracts had data from many more people. This work was reviewed and approved by Microsoft's ethics review board and Institutional Review Board (IRB).

## Landscape features

To incorporate landscape features in our example "in practice" model, we geo-referenced the NLCD data using the boundaries of census tracts and calculated the proportion of area of a census tract covered by each land cover sub-class. Given that these proportions can be similar across multiple census tracts despite the underlying distribution of land cover being different (e.g., two census tracts can have the same percentage of highly developed land, but that land can be geographically concentrated in one tract and distributed in the other), we introduced additional features to represent the distribution of land cover features. For each census tract we calculate features representing i) the proportion of high developed land pixels neighboring another high developed land pixel, ii) the proportion of high developed land pixels neighboring a forest pixel, and iii) the proportion of forest pixels neighboring another forest pixel. These features were intended to capture the notion that developed land corresponds to higher level of pollution while forests and trees can be a source of pollen, both of which have adverse impact on patients suffering from asthma and COPD.

As indicated, individuals are not restricted to only the census tract they live in. They travel for work and recreational purposes and experience environmental factors of neighboring areas [33]. To incorporate these affect of neighbouring census tracts, similar to search queries, for each census tract, we average the proportions of landscape features from $m$ closest census tracts.

## Model

We use an ensemble model based on a random forest algorithm to generate estimates of respiratory illnesses. Random forest models are known for their tolerance to overfitting on training data [34]. This makes them ideal for application like ours, where due to the small size of spatial units, there can be outliers in the predictors. In some cases we introduce a state specific term in our feature set to allow for incorporation of baseline prevalence of respiratory illnesses in each state, and note in the results which models include this term.

We used 5-fold cross validation and test the performance of the model on the out of sample held out folds, using two metrics of our predictions in relation to ground truth as determined by the CDC: correlation of predicted and actual illness rates and mean absolute error (MAE) of illness rates. The mean absolute error helps us to determine the average error in each prediction while the correlation metric quantifies how well each model captures relative levels of respiratory illness prevalence across census tracts.

We note here that we tested additional models, including linear regression, logistic regression, and support vector machines (SVM), but the random forest model consistently provided the best results in every setting and hence was chosen for the study. Given, that the aim of the work was to identify the utility of search query data in the estimation of the geographical distribution of disease, as opposed to comparing the predictive capabilities of machine learning models, we report only the random forest model results.

## Census tract and neighboring tract selection

The CDC dataset consisted of a total of 29305 census tracts. We filtered out any census tracts from the dataset which had either the census or land cover data missing. This was done to ensure fair comparison between models. In addition, we also excluded census tracts from which there were fewer than 50 distinct users contributing queries to our sample, to avoid privacy concerns. This left us with a total of 23897 census tracts for training and testing models. The census tracts had a median area of 2.45 $km^2$ (min = 0.014, max = 384.7) and a median population of 3652 (min = 50, max = 28960). Fig 1 shows the frequency distribution of area and population across the census tracts highlighting the small geographical granularity at which the study was performed.

To identify the appropriate effect of incorporating neighboring census tracts into the prediction task, we explore the performance of models trained on data generated using $m$ = 1, 4, 8, 16 and 32 neighboring tracts. Fig 2 shows the correlation between predicted and actual prevalence of asthma and COPD, on training data, for various values of $m$ for search queries and land cover based models. As evident from the figure, the value of $m$ = 8 provides the highest correlation for both asthma and COPD when using search query data and for COPD when using land cover data. For land cover data and the asthma outcome measure (Fig 2(C)) the optimized value of $m$ was 16, but given that the correlation value for $m$ = 8 was close, for consistency we use the values of $m$ = 8 throughout the analysis. Socioeconomic and demographic measures are strictly defined with the boundary of each tract, and thus for census data we do not incorporate the affect of neighboring census tracts.

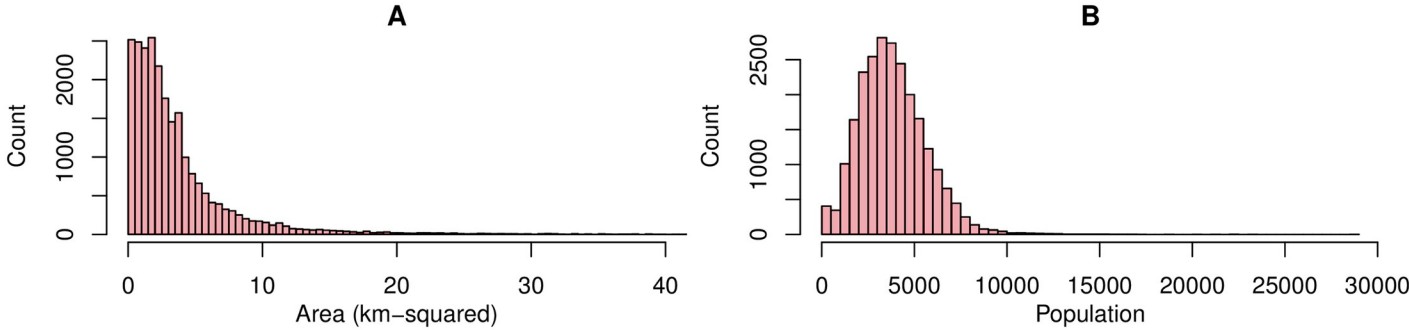

**Fig 1. Distribution of area and population of census tracts.** A: Frequency distribution of area in kilometer squared and B: population of census tracts included in the study.

## Results

### Model performance

We start our analysis by answering the simplest question: How well can search query data alone, without any other data sources or features related to location, estimate the geographical distribution of respiratory illnesses, at a census tract level, across the U.S.? When tested against

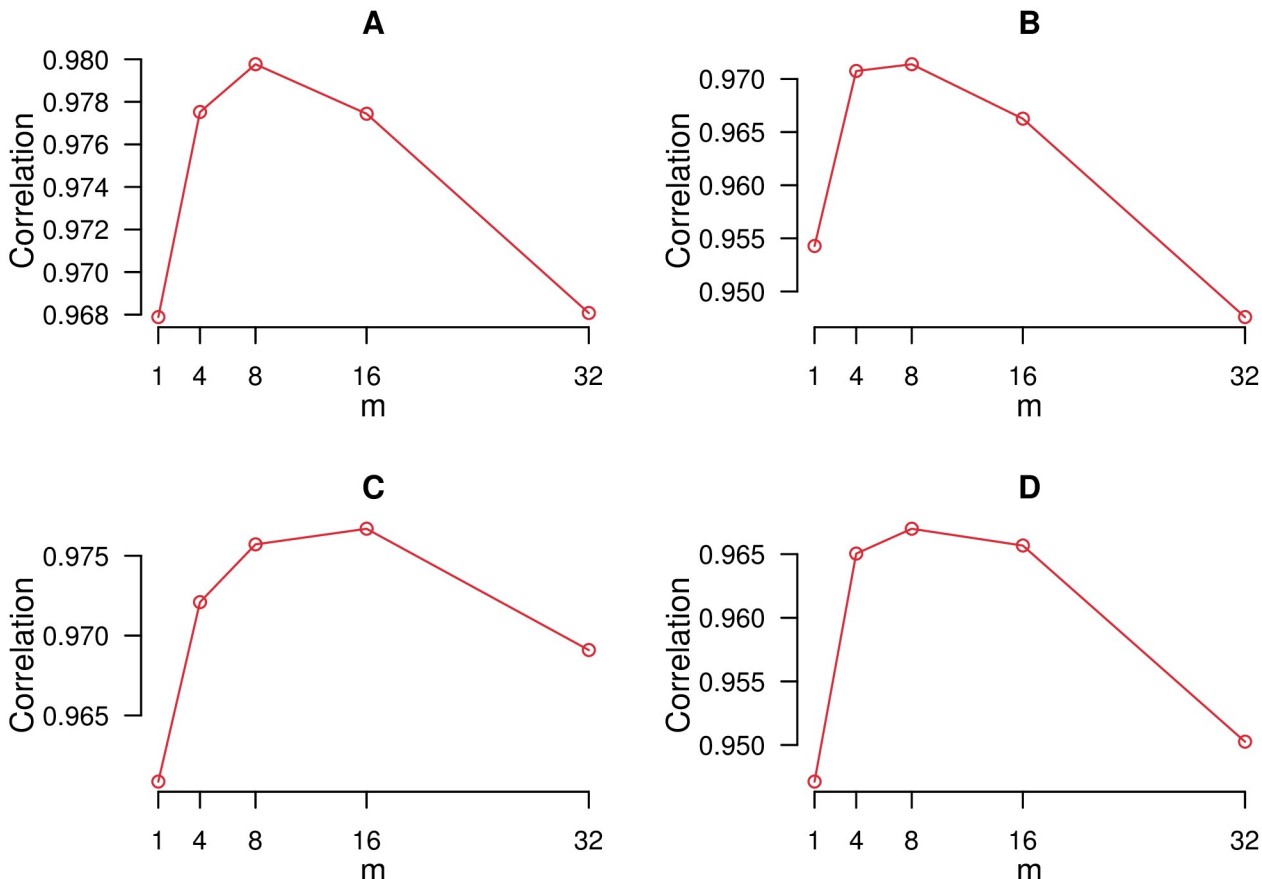

**Fig 2. Correlation of prediction on training data against the number of nearby census tracts used to perform prediction ($m$).** A: Search query based model for asthma. B: Search query based model for COPD. C: Land cover model for asthma. D: Land cover model for COPD.

hold-out data, our model trained on only search query data provided a good fit with the ground truth rate of asthma and COPD, providing a predicted to actual illness rate correlation of 0.755 and MAE of 1.012 for asthma, and correlation of 0.685 and MAE of 1.384 for COPD.

To incorporate variations in prevalence of respiratory illnesses across different states in US, we then added a slight enhancement to the base search query data model by incorporating an additional feature of which state a census tract belongs to. Variations such a coding might capture include any combination of baseline prevalence of respiratory illnesses, variation in internet usage trends, or variations in socioeconomic, demographic, and environmental factors. Including this simple feature in the model improved our estimates, providing a predicted to actual illness rate correlation of 0.838 and MAE of 0.804 for asthma, and a correlation of 0.744 and MAE of 1.236 for COPD, on the held out folds during cross-validation.

Next we explore how the addition of search query data to other data sources improves our estimates of illness rates, illustrating potential use cases in practice. Adding landscape features to our search query data model increased the correlation between predicted and actual illness rates for asthma to 0.854 and for COPD to 0.766. Similarly, MAE dropped for asthma to 0.768 and for COPD to 1.190.

Finally we train a single "best" model that incorporates all data sources: search queries, land cover, and census data. This model captures the ideal situation in which land cover and census data are as readily available as search query data, and reflects what could be done by practitioners seeking to achieve maximum predictive performance. This model indeed provides the best performance on the held out folds during cross validation, providing a predicted to actual illness rate correlation of 0.907 and MAE of 0.628 for asthma and a correlation of 0.875 and MAE of 0.882 for COPD. To put the MAE values in context, the average asthma prevalence reported across the census tracts in the U.S was 9.783%, and hence the average forecast would be well within 1% of that in absolute terms. Similarly, for COPD the average prevalence of the illness was 6.416%, and the average forecast would also be well within 1% of that in absolute terms. Table 2 summarizes the correlation and MAE values for different models based on different data sets.

Figs 3 and 4 show the predicted and actual prevalence of asthma and COPD for the census data based model (A), search query based model (B), and combined model based on census, land cover and search query data (C). As evident from the figure (data points concentrated around $y = x$ line) our models provide generally good census tract level estimates of the prevalence of respiratory illnesses as reported by the CDC. Though most predictions are close to actual CDC values, for COPD and for census tracts with the highest reported prevalence, the performance of the models decreases. Finally, as evident from the figure, the model based on all data sources provides the best predictions of respiratory illness prevalence.

As a concrete example, Figs 5 and 6 show the CDC estimates (A), search query based model prediction (B), and all data sources based model prediction (C) values of prevalence of asthma and COPD for the city of Chicago. The city of Chicago was chosen as an example given that it has one of the highest number of census tracts across cities in our dataset and has a high variation in prevalence of both asthma and COPD.

**Table 2. Correlation and MAE values for models trained using different data sources.**

|  | Metrics | Search | Search w/ State | Search & Land | Search, Land & Census |
|---|---|---|---|---|---|
| Asthma | Correlation | 0.755 | 0.838 | 0.854 | 0.907 |
|  | MAE | 1.012 | 0.804 | 0.768 | 0.628 |
| COPD | Correlation | 0.685 | 0.744 | 0.766 | 0.875 |
|  | MAE | 1.384 | 1.236 | 1.190 | 0.882 |

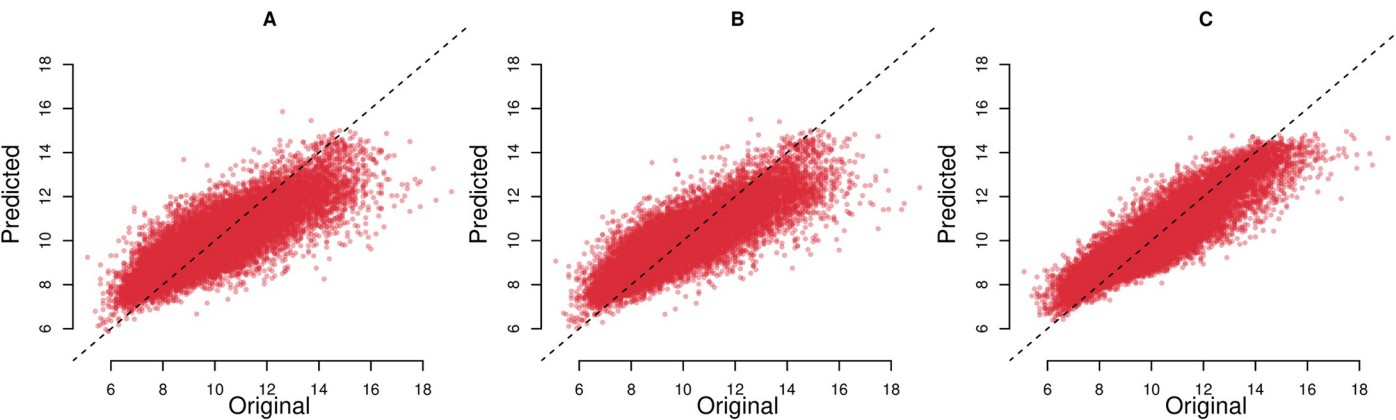

**Fig 3. Predicted against original prevalence of asthma.** A: Model based on search queries. B: Model based on search queries and land cover data. C: Model based on search queries, census and land cover data. Each point denotes a distinct census tract in the dataset. Dotted line represent x = y.

## Top predictors

We also report the top predictive features from search query data when generating estimates of respiratory illnesses. These features are based on the node purity values of features in the random forest model. Table 3 shows the top 5 features from COPD and asthma prevalence models. We find the smoking alternative related terms 'juul' and 'vape' to be amongst the top predictors. For COPD, while the term 'copd' was the second most predictive features, interestingly we do not find the term 'asthma' to be a top predictor of the prevalence of asthma. Other related drug terms such as incruse and ketamine were also amongst the top predictors.

## Discussion

Analyses such as these that deal with fine-grained geographic aggregations, often face methodological issues related to data sparsity in a given region. We employed several techniques to handle this with search query data.

To start, we expand the set of topically relevant keywords from which to identify matching search queries empirically using a word embedding approach. The decision to use this approach, as opposed to looking for correlations among potential keywords as has been used

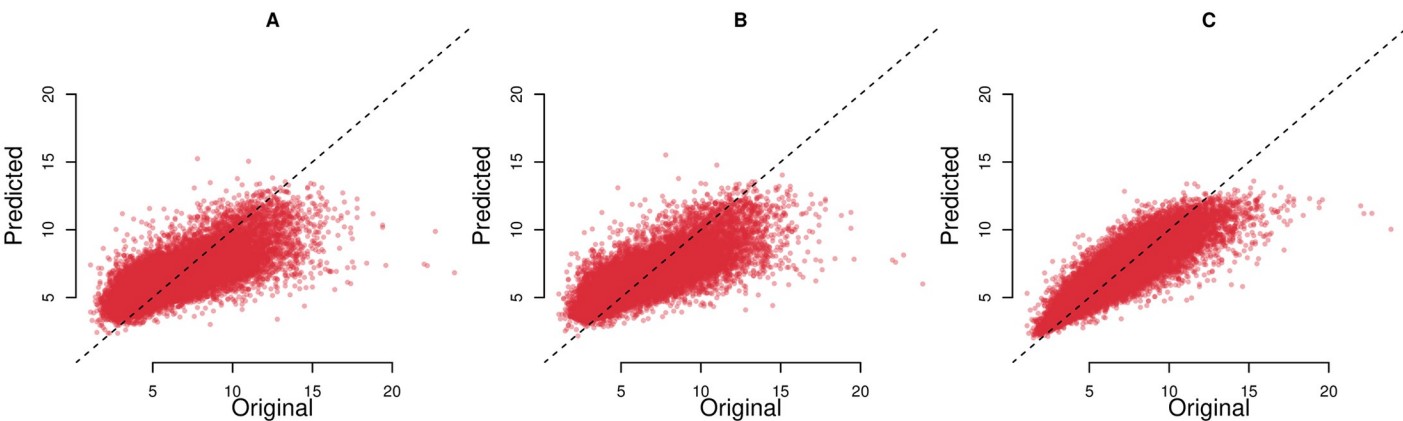

**Fig 4. Predicted against original prevalence of COPD.** A: Model based on search queries. B: Model based on search queries and land cover data. C: Model based on search queries, census and land cover data. Each point denotes a distinct census tract in the dataset. Dotted line represent x = y.

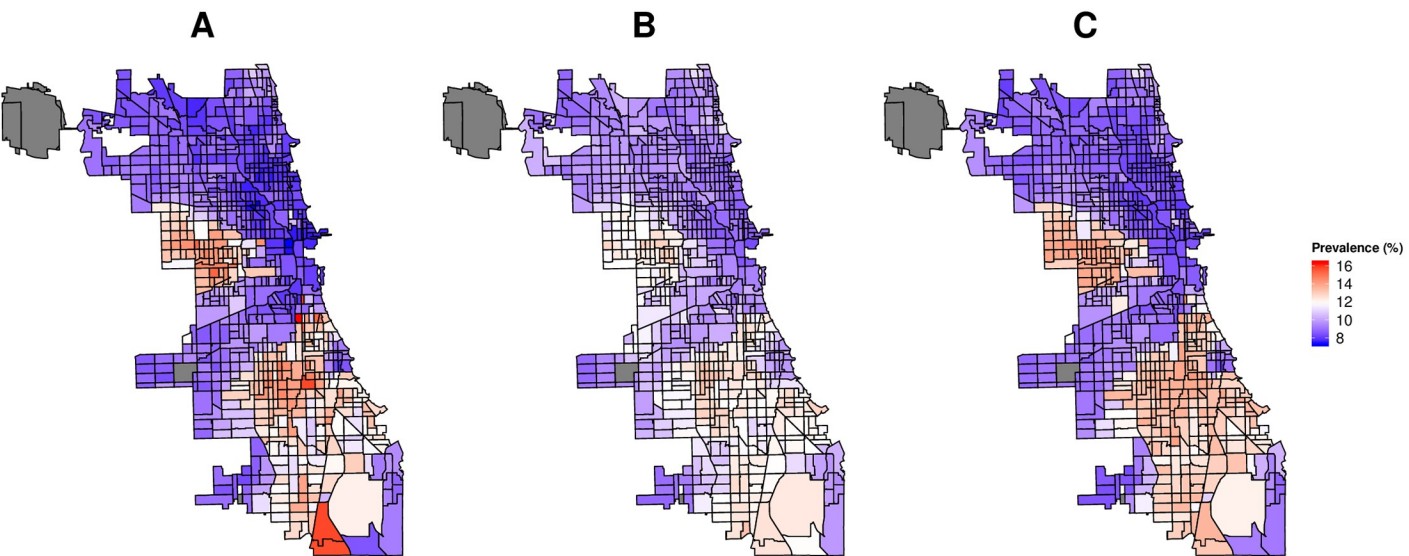

**Fig 5. Census tract level prevalence of asthma for the city of Chicago in U.S.** A: Ground truth as estimated by CDC. B: Estimates from search query based model. C: Estimates from model based on search queries, land cover and census data. Greyed areas show census tracts for which data was not available from any one of the data sources.

in previous work, was made for several reasons. First, in contrast to previous work that looks at disease variation over time and leverages seasonal patterns, our work focuses on the prevalence of non seasonal diseases at small geographical units. The lack of temporal component makes it difficult to identify terms using a correlation analysis. Even search query based tools like 'Google Trends' [35] are designed for reporting correlation in time rather than space.

Looking for terms that are correlated without any temporal component means that they must be correlated either geographically (originating from the same location) or within the

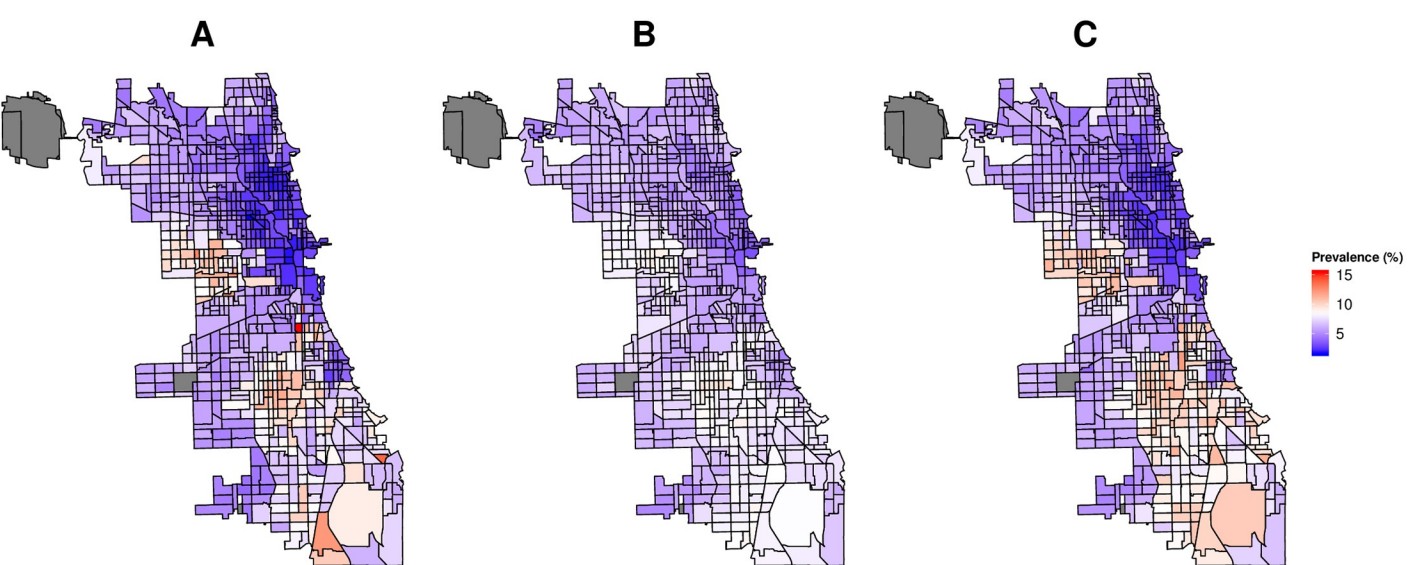

**Fig 6. Census tract level prevalence of COPD for the city of Chicago in U.S.** A: Ground truth as estimated by CDC. B: Estimates from search query based model. C: Estimates from model based on search queries, land cover and census data. Greyed areas show census tracts for which data was not available from any one of the data sources.

**Table 3. Top 5 predictive search query terms for asthma and COPD models.**

|   | Asthma | COPD |
|---|--------|------|
| 1 | juul | juul |
| 2 | vape | copd |
| 3 | migraine | vape |
| 4 | incruse | ketamine |
| 5 | copd | tobacco |

text corpus (cooccurring in the same query). The word embedding approach allowed us to use a national-scale sample of queries, avoiding sparsity due to small geographic regions, and to avoid a term cooccurrence requirement. Avoiding such a cooccurrence requirement in favor of identifying terms that occur in similar word contexts is more aligned with search behavior in that people are likely to use related terms similarly in a search, but not include both of the terms. Finally, the word embedding approach is empirical in that it does not require that we specify in advance a set of potentially related terms.

A second technique we utilized to handle data sparsity was to incorporate search query features from neighboring census tracts, showing that the closest eight tracts was optimal for generating illness rates. Again this leverages natural human behavior in that people are likely to visit and be affected by adjacent census tracts and thus signals from those census tracts are relevant. This also had the benefit of smoothing over some level of uncertainty with the reverse IP lookup. This approach of accounting for uncertainty by averaging information from nearby locations has been used in the past to predict localized poverty levels [26].

Regarding use of land cover data, as opposed to using a deep learning framework to identify segmentation maps of land cover features, we used pre-classified land cover maps from NLCD. This decision was made due to the large scale of our study where we needed high resolution land cover maps for the entire U.S. As mentioned, deep learning semantic segmentation models trained on data from one location often do not provide good prediction in other locations [28]. Thus in absence of a land cover segmentation training dataset from locations across the entire U.S., it would be nearly impossible to generate reliable estimates of land cover maps.

Turning to the results of the work, we first showed that a model based solely on search query data, and then one that added a simple state specific coding feature could generate out of sample predictions that correlated in the 0.7 to 0.8 range with actual illness rates. The decision to add only the state coding as an additional feature was made to ensure that the model can be generalized to estimate prevalence of respiratory illnesses in other parts of U.S for which no ground truth data was available. Adding a city or county specific feature potentially could improve the estimates, as they provide estimation of location specific information at a higher resolution, but doing so would make the model inapplicable to cities not included in 500 cities project.

In practice, we believe that, despite search query data alone providing good estimation of neighborhood level respiratory illness rates, additional data sources such as landscape features and census tracts can be added to the model to further improve the prediction. This is evident from the results shown in Table 2 where addition of landscape and census data further improved the estimation of neighborhood level respiratory illness rates. Thus, given the practicality and advantage of using these additional features, here we have also discussed simple ways on how such features can be added into search query based models.

In terms of those identified keyword features for the search query based model, we find that vape related terms i.e 'vape' and 'juul' (a popular electronic cigarette brand) were the most

predictive features for both asthma and COPD. Vapes and electronic cigarettes are alternative forms of nicotine delivery used by adults trying to quit smoking. This may suggest that an increased number of people in census tracts with higher rates of respiratory illness are trying to find alternative forms of nicotine delivery systems. Medication names such as 'incruse' and 'ketamine' were also highly predictive of respiratory illness prevalence, reinforcing our strategy to use medication names as features in our model. One observation from the top predictors is that while the term 'copd' is highly predictive of COPD prevalence in a census tract, the term 'asthma' is not amongst the top predictors of asthma. Potentially this reflects the fact that asthma patients often develop the disease at an early stage in life and thus less likely to search for it directly compared to COPD, which develops later in life.

Finally, we discuss the trade-offs of using individual data sources for generating census tract level estimates. Census data is collected by the government as part of a larger data effort that costs billions of dollars to collect. The cost of U.S census survey of 2020 was estimated to be approximately 15.6 billion dollars [36]. Given the cost of data collection, the data is collected roughly every five years and estimates for remaining years are generated using American Community Surveys (ACS). Further given the small geographical scale of census tracts, the census bureau reports census level information as averages of 5 years. Similarly, the NLCD land cover data is published at an interval of several years and is backdated: the last two published datasets were for year 2013 ans 2016 and the data for 2016 was published in early 2019. In contrast search query data may be available much closer to contemporaneous with present day and could be used for more frequent updating of estimates of respiratory illnesses. As indicated by our best model, we recommend combining census and land cover data with search query data to provide more reliable and accurate estimates of prevalence of respiratory illnesses.

## Limitations and future work

As with any search query related data, the approach will only work in areas with sufficient internet penetration. The number of internet users in a location can depend on several factors including education level, income, and employment type. In our example 'in practice' model, which incorporates census data, we include levels of poverty, which can capture variations in internet usage, but only as a rough proxy. We note that every census tract in our study included a minimum of 50 unique people searching for the illness related terms, and a clear majority had far more than that, suggesting reasonable sampling properties, but we do not know the actual demographics of the people contributing queries.

Second, our approach is designed for non-infectious diseases and if applied to infectious diseases might underestimate the prevalence of disease. This is due to the fact that prevalence of infectious diseases is known to be dependent on the prevalence of disease in neighboring areas [37]. In our work, we do incorporate trends of search queries from neighboring census tracts, but further studies should quantify the exact effect of trends of search queries from neighboring census tracts in a given census tract. It should be noted that these effects only arise when the prevalence of disease is studied at a fine sub-city spatial granularity and thus haven't been discussed in previous search query based studies as they deal with predicting the prevalence of disease at coarser geographical units.

Finally, we comment on the availability of query data. For privacy reasons, search engine companies cannot, and should not, ever share raw query data with the public. However, as mentioned, we envision a scenario in which geographically aggregated and anonymous indices could be made available to the community of researchers and public health practitioners for research, validation, and application purposes. Roughly this could work similar to the recent

contribution by Facebook of maps showing displacement patterns following natural disasters that have been shared with public health researchers in a privacy preserving manner [38].

To the best of our knowledge, this work is the first to leverage signals from readily available search query data to infer localized census tract level prevalence of illnesses. Though the application is shown for respiratory illnesses, we believe that our approach will open venue for researchers to study the prevalence of other diseases at fine-grained sub-city level spatial units.

## Author Contributions

**Conceptualization:** Nabeel Abdur Rehman, Scott Counts.

**Data curation:** Nabeel Abdur Rehman, Scott Counts.

**Formal analysis:** Nabeel Abdur Rehman, Scott Counts.

**Investigation:** Nabeel Abdur Rehman, Scott Counts.

**Methodology:** Nabeel Abdur Rehman, Scott Counts.

**Supervision:** Scott Counts.

**Validation:** Nabeel Abdur Rehman, Scott Counts.

**Visualization:** Nabeel Abdur Rehman, Scott Counts.

**Writing – original draft:** Nabeel Abdur Rehman, Scott Counts.

**Writing – review & editing:** Nabeel Abdur Rehman, Scott Counts.

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
