## [Decision Letter · Decision Letter 0]

19 Apr 2021

PONE-D-21-05880

Neighborhood level Respiratory Illnesses Rate Estimation Using Search Query Data

PLOS ONE

Dear Dr. Abdur Rehman,

Thank you for submitting your manuscript to PLOS ONE. After careful consideration, we feel that it has merit but does not fully meet PLOS ONE’s publication criteria as it currently stands. Therefore, we invite you to submit a revised version of the manuscript that addresses the points raised during the review process.

 Authors are requested to address the comments of reviewer 1. 

We look forward to receiving your revised manuscript.

Kind regards,

Usman Qamar

Academic Editor

PLOS ONE

Journal Requirements:

4. Thank you for stating the following in the Financial Disclosure section:

'The author(s) received no specific funding for this work.'

We note that one or more of the authors are employed by a commercial company: Microsoft Research

Reviewers' comments:

Reviewer's Responses to Questions

**Comments to the Author**

1. Is the manuscript technically sound, and do the data support the conclusions?

Reviewer #1: Yes

2. Has the statistical analysis been performed appropriately and rigorously? 

Reviewer #1: Yes

3. Have the authors made all data underlying the findings in their manuscript fully available?

Reviewer #1: Yes

4. Is the manuscript presented in an intelligible fashion and written in standard English?

Reviewer #1: Yes

5. Review Comments to the Author

Reviewer #1: 1. I recommend changing the terms in the title to chronic respiratory disease prevalence estimation. Respiratory illness is too broad, as it would include influenza, COVID-19, and other infectious diseases that were not examined. And prevalence is the correct epidemiology term for what the authors are estimating.

2. The authors report a correlation of 0.91 in the abstract, but in the results this only relates to a full model with the outcome of asthma. Can the authors clarify in this sentence what measure of correlation is this 0.91 and also report correlations from their other outcomes? It’s probably not necessary to report all the models, just the full models with the best correlations.

3. I recommend removing the last sentence of the abstract. This sentence does not add anything to the article, and is only a claim of “first”. Let the science speak for itself.

4. In the introduction the authors claim COPD to be the third leading cause of death in the USA. From the CDC website the third leading cause of death in the USA is accidents, and lower and chronic lower respiratory diseases are fourth (not just COPD alone). There is also an anomaly with the pandemic where COVID-19 was the third leading cause of death in 2020 and perhaps will be as high in 2021. I recommend revising this introduction as follows: “… more than 15.7 million people suffer from Chronic Obstructive Plumonary Disease (COPD). Chronic lower respiratory diseases, including asthma and COPD, are a leading cause of death in the USA, accounting for an estimated X% of all-cause mortality each year.”

6. PLOS authors have the option to publish the peer review history of their article (what does this mean?). If published, this will include your full peer review and any attached files.

Reviewer #1: No

---

## [Author Response · Author response to Decision Letter 0]

14 May 2021

Reviewer #1: I recommend changing the terms in the title to chronic respiratory disease prevalence estimation. Respiratory illness is too broad, as it would include influenza, COVID-19, and other infectious diseases that were not examined. And prevalence is the correct epidemiology term for what the authors are estimating.

We thank the reviewer for their suggestion and have updated the title of our study to reflect correct terminologies. The new title is “Neighborhood level chronic respiratory disease prevalence estimation using search query data”.

Reviewer #1: The authors report a correlation of 0.91 in the abstract, but in the results this only relates to a full model with the outcome of asthma. Can the authors clarify in this sentence what measure of correlation is this 0.91 and also report correlations from their other outcomes? It’s probably not necessary to report all the models, just the full models with the best correlations.

We have added updated the abstract to reflect the performance of models for both asthma and COPD.

Reviewer #1: I recommend removing the last sentence of the abstract. This sentence does not add anything to the article, and is only a claim of “first”. Let the science speak for itself.

We agree with the reviewer and have removed the sentence from the abstract.

Reviewer #1: In the introduction the authors claim COPD to be the third leading cause of death in the USA. From the CDC website the third leading cause of death in the USA is accidents, and lower and chronic lower respiratory diseases are fourth (not just COPD alone). There is also an anomaly with the pandemic where COVID-19 was the third leading cause of death in 2020 and perhaps will be as high in 2021. I recommend revising this introduction as follows: “… more than 15.7 million people suffer from Chronic Obstructive Plumonary Disease (COPD). Chronic lower respiratory diseases, including asthma and COPD, are a leading cause of death in the USA, accounting for an estimated X% of all-cause mortality each year.”.

We thank the reviewer for pointing this out and have updated the introduction to reflect up to date information.

---

## [Editor Report · Decision Letter 1]

17 May 2021

Neighborhood level chronic respiratory disease prevalence estimation using search query data

PONE-D-21-05880R1

Dear Dr. Abdur Rehman,

We’re pleased to inform you that your manuscript has been judged scientifically suitable for publication and will be formally accepted for publication once it meets all outstanding technical requirements.

Kind regards,

Usman Qamar

Academic Editor

PLOS ONE
---

## [Editor Report · Acceptance letter]

31 May 2021

PONE-D-21-05880R1 

Neighborhood level chronic respiratory disease prevalence estimation using search query data 

Dear Dr. Abdur Rehman:

I'm pleased to inform you that your manuscript has been deemed suitable for publication in PLOS ONE. Congratulations! Your manuscript is now with our production department. 

Kind regards, 

on behalf of

Dr. Usman Qamar 

Academic Editor

PLOS ONE